# Characterization and Evaluation of Bone-Derived Nanoparticles as a Novel pH-Responsive Carrier for Delivery of Doxorubicin into Breast Cancer Cells

**DOI:** 10.3390/ijms21186721

**Published:** 2020-09-14

**Authors:** Sheikh Tanzina Haque, Rowshan Ara Islam, Siew Hua Gan, Ezharul Hoque Chowdhury

**Affiliations:** 1Jeffrey Cheah School of Medicine and Health Sciences, Monash University Malaysia, Jalan Lagoon Selatan, Bandar Sunway, Subang Jaya 47500, Selangor, Malaysia; sheikh.haque@monash.edu (S.T.H.); rowshan.islam@monash.edu (R.A.I.); 2School of Pharmacy, Monash University Malaysia, Jalan Lagoon Selatan, Bandar Sunway, Subang Jaya 47500, Selangor, Malaysia; Gan.SiewHua@monash.edu; 3Tropical Medicine and Biology Multidisciplinary Platform, Monash University Malaysia, Jalan Lagoon Selatan, Bandar Sunway, Subang Jaya 47500, Selangor, Malaysia

**Keywords:** goose bone ash (GBA), carbonated apatite, breast cancer, carbonate apatite (CA), pH responsive drug delivery, cytotoxicity, cellular uptake, DOX binding, protein corona, nano-carrier

## Abstract

**Background**: The limitations of conventional treatment modalities in cancer, especially in breast cancer, facilitated the necessity for developing a safer drug delivery system (DDS). Inorganic nano-carriers based on calcium phosphates such as hydroxyapatite (HA) and carbonate apatite (CA) have gained attention due to their biocompatibility, reduced toxicity, and improved therapeutic efficacy. **Methods**: In this study, the potential of goose bone ash (GBA), a natural derivative of HA or CA, was exploited as a pH-responsive carrier to successfully deliver doxorubicin (DOX), an anthracycline drug into breast cancer cells (e.g., MCF-7 and MDA-MB-231 cells). GBA in either pristine form or in suspension was characterized in terms of size, morphology, functional groups, cellular internalization, cytotoxicity, pH-responsive drug (DOX) release, and protein corona analysis. **Results**: The pH-responsive drug release study demonstrated the prompt release of DOX from GBA through its disintegration in acidic pH (5.5–6.5), which mimics the pH of the endosomal and lysosomal compartments as well as the stability of GBA in physiological pH (pH 7.5). The result of DOX binding with GBA indicated an increment in binding affinity with increasing concentrations of DOX. Cell viability and cytotoxicity analysis showed no innate toxicity of GBA particles. Both qualitative and quantitative cellular uptake analysis in both cell lines displayed an enhanced cellular internalization of DOX-loaded GBA compared to free DOX molecules. The protein corona spontaneously formed on the surface of GBA particles exhibited its affinity toward transport proteins, structural proteins, and a few other selective proteins. The adsorption of transport proteins could extend the circulation half-life in biological environment and increase the accumulation of the drug-loaded NPs through the enhanced permeability and retention (EPR) effect at the tumor site. Conclusion: These findings highlight the potential of GBA as a DDS to successfully deliver therapeutics into breast cancer cells.

## 1. Introduction

Although conventional chemotherapeutics has been a major impetus for breast cancer treatment, its adverse effects tend to limit their potentials. The drugs normally have to be administered in high doses to maintain a high systemic concentration, thus rendering undesirable toxic effects. Therefore, the development of safe therapeutic modalities of higher efficacy is of prime importance. The limitations associated with conventional drugs can be surmounted through the use of nanotechnology [1], which offers major advantages in treating cancer through site-specific drug delivery.

Nanoparticle-based drug delivery systems (NPDDSs) have emerged as competent carriers for therapeutic cargos. They can (1) prolong circulation by increasing the uptake at the tumor site through the enhanced permeability and retention (EPR) effect, (2) provide tunable shape and size, (3) overcome multi-drug resistance (MDR) by avoiding p-glycoprotein recognition, (4) provide controlled release of drugs by different stimulus (e.g., pH, temperature, enzymes, biological or chemical agents), (5) prevent drug degradation, and (6) increase drug uptake in cancer cells [2,3,4,5,6]. However, the intravenous administration of NPDDS poses several obstacles. First, they undergo opsonization, which results in non-specific accumulation of nanoparticles (NPs) into the spleen and the liver through the action of macrophages of the mononuclear phagocyte system. Secondly, the NPs can be expelled from the tissues due to interstitial fluid pressure [7]. In the past few decades, stimuli (pH, temperature, light, enzymes, electric field, magnetic field, redox potential, and biochemical trigger [8,9,10,11,12,13,14]) responsive nano-carriers have received special attention for their application in DDSs. Among the various different stimuli, the pH-responsive nature of NPs is indispensable. For the precise targeting of mammary cancer, pH-responsive NPDDS function by responding only to the acidic pH of endosomes (pH 5.5–6) or lysosomes (pH 4.5–5) to release the encapsulated therapeutic cargo in the cytosol (pH 7.4) of the cancer cells, while remaining stable at physiological pH (pH 7.4). That is, pH-dependent NPDDSs are sensitive to the pH of the diseased cells. In addition, pH-dependent NPDDS are considered optimal, since they take advantage of the EPR effect to deliver drugs via passive targeting at the tumor site [15,16,17]. However, existing pH-dependent NPDDSs (e.g., lipids, polymers, and inorganic NPs [18,19]) face several challenges with regard to their route of delivery, selection of suitable materials, fabrication methods, and modification of properties in order to prevent premature drug degradation [16]. For example, pH-responsive liposomes [20] pose certain limitations such as poor drug loading efficiency, instability, low blood circulation half-life, sterilization issues, poor scalability, and the rapid release as well as degradation of drugs [21,22]. pH-sensitive polymeric micelles [23] and hydrogel [24] usually suffer from reduced biodegradability, low mechanical strength, and mitigated drug release, which are especially true with micelles and polymers containing hydrazone, orthoester, ketal, and acetal functional groups in an acidic pH [18]. On the other hand, pH-dependent inorganic NPs of carbonate apatite (CA) and calcium carbonate (CaCO_3_), for example, have the tendency to self-aggregate and might lead to long-term toxicity and rapid clearance before reaching the site of action [22,25]. Therefore, it is of paramount importance to develop a new pH-responsive nano-carrier system with hallmarks including biodegradability, bioavailability, enhanced drug loading efficacy, and the prevention of premature drug leaking before the particles reach the site of action.

Nevertheless, the behavior of the NPDDS in a biological environment is complicated. For example, when the engineered NPs are introduced into a biological environment, serum proteins immediately adsorb on the surface of the NPs, resulting in the generation of a biological layer known as “protein corona”, which can modify the biological identity of the NPs by changing their morphology, charges, nature of aggregation, and interfacial composition. These modifications affect NPs’ biodistribution, degradation, clearance, cellular uptake, and toxicity. As a result, the adsorbed proteins can either hinder or increase the uptake of NPs inside cells (i.e., by interacting with the cognate receptors present on cell membranes) [26,27,28,29,30]. Therefore, it is important to investigate how blood proteins adhere to NPs to determine the fate of the NPs once inside the body.

To date, inorganic nano-carriers, especially those based on calcium phosphates such as hydroxyapatite and CA, have been intensively investigated as carriers for drugs due to their good biocompatibility, low toxicity, and enhanced efficacies [31,32,33,34,35]. An interesting candidate is bone, which is natural and can be processed into bone ash by calcination or into charcoal by burning with or without oxygen. Goose bone ash (GBA) or bone charcoal is a form of CA and/or carbonated apatite [36]. Due to the porous structure and its adsorption capacity, it is used as a water-purifying agent to remove a wide range of minerals including fluorine, copper, cadmium, and lead from water [37,38]. Moreover, the conditions for the production of GBA are cost-effective, simple, and attainable. In addition, the partial substitution of carbonate (CO_3_^2−^) for phosphate (PO_4_^3^) and sodium (Na^+^) for calcium (Ca^2+^) in GBA may occur. These substitutions are similar to bone mineral [39,40]. Furthermore, the generation of GBA particles will be reproducible in the context of size distribution as opposed to other synthetic NPs, which vary dramatically with a small change in different parameters (pH, temperature, coating agents).

To date, CA has already received attention as a promising delivery system for drugs and genetic materials in pre-clinical models [41,42]. In this article, GBA, a natural derivative of CA or carbonated apatite, is characterized and investigated as a potential delivery system for doxorubicin (DOX). MCF-7 and MDA-MB-231 human breast adenocarcinoma cell lines were exploited in in vitro models. DOX is selected as a model drug, since it is the most effective anthracycline extensively used in breast cancer due to its ability to intercalate DNA, prevent DNA replication, and release free radicals, thus inhibiting protein function and eventually leading to the apoptosis of cancer cells. The use of DOX is often restricted due to the presence of serious adverse events such as nausea, vomiting, and heart failure [43,44]. DOX may act with GBA via electrostatic interaction in which its positive charge reacts with CO_3_^2−^ and PO_4_^3−^, which are negatively charged. In a previous study [45], it was revealed that GBA is non-toxic when administered in vivo. To the best of our knowledge, no research has been reported on GBA’s utility in the successful delivery of drugs into cancer cells.

## 2. Results and Discussion

### 2.1. Verification of Particle Aggregation through Optical Micrographs and Turbidity Measurement

Microscopic images were used to analyze details regarding the aggregation of different concentrations of GBA suspension in DMEM at pH 7.5 and 37 °C (Appendix A). All images were taken at a magnification of 10× (scale bar of 50 µm). Large aggregates were observed for 0.1 mg/mL GBA suspension, whereas significantly fewer aggregates were detected for 0.01 mg/mL GBA suspension. Interestingly, diluted concentrations (e.g., 0.01, 0.10, and 1.00 µg/mL) of GBA suspension exhibited considerably fewer aggregates. Thus, it is plausible that large aggregates in high GBA concentration can be mitigated by lowering the concentration of GBA suspension.

The aggregation of GBA particles was also verified through turbidity measurement by means of a UV-VIS spectrophotometer (1800 MS; 320 nm) (Table 1). An increment in turbidity was observed due to the aggregation of the particles with increasing concentrations (0.01, 0.10, 1.00, 10, and 100 mg/mL) of GBA in water (Table 1). For example, 0.01, 0.10, 1.00, 10, and 100 mg/mL of GBA in water demonstrated a turbidity of around 0.27, 0.73, 2.07, 3.24, and 3.72, respectively. The reduced aggregation of the particles at lower concentrations might be due to the dilution with water or the addition of fetal bovine serum (FBS), which resulted in smaller, less aggregated particles compared to the particles at higher concentrations.

### 2.2. Particle Size Measurement of GBA Suspension by Dynamic Light Scattering (DLS) Method

Size is the most critical parameter for an effective nano-carrier, as it affects the stability, cellular internalization, and the pathways of cellular internalization for the drug-loaded nano-carriers [46]. For example, NPs with size greater than 200 nm can enter the cancer cell through the clathrin-mediated pathway [47]. Moreover, size can influence the self-aggregation between the NPs [48] and modulate the removal and distribution of the drug-loaded nano-carriers in different tissues in vivo [46]. Particles with a hydrodynamic diameter of less than 8 nm are excreted from the body through renal (kidney) clearance, whereas particles with a hydrodynamic diameter of greater than 1 µm can be engulfed by the macrophages of MPS in the spleen, bone marrow, and liver [40,46,49]. Furthermore, the size of NPs can influence the composition of the protein corona formed on their surface, which can modulate their distribution in different tissues and cellular internalization by altering the composition, thickness, and activities of proteins [50,51].

The z-average diameters of GBA suspended in different concentrations (1 μg/mL, 0.01 mg/mL, 0.10 mg/mL, 1.00 mg/mL, 10.00 mg/mL, and 100.00 mg/mL) in water were measured after an incubation step (37 °C for 30 min). The step was followed with the addition of FBS (10%). A dramatic increase in particle size was observed with increasing concentrations of GBA in water (Table 2). The reduction in size at lower concentrations might be due to the dilution in water or the addition in FBS, which significantly mitigated the aggregation between them and resulted in smaller particles. These results were similar to those of the turbidity analysis (Table 1) and optical microscopic images (Appendix A). Representative images for particle size distribution by intensity are demonstrated in Appendix A.

### 2.3. Morphological Analysis of GBA Suspension by FESEM

The FESEM micrographs of GBA suspension (1 mg/mL) showed that GBA particles have spherical morphology. These images revealed that a substantial number of particles were smaller with an average particle diameter of 12–20 nm. Nevertheless, fewer particles displayed an average particle diameter of approximately 100 to 120 nm (Figure 1A). However, GBA suspension at 10 mg/mL produced irregular-shaped crystals in clustered forms. Therefore, a higher concentration of GBA suspension displayed more crystalline structure compared to lower concentration. The shape and sizes of the GBA crystals were concentration-dependent, demonstrating bigger and more aggregated particles at higher concentrations. The average particle diameter of the particles in 10 mg/mL GBA suspension was approximately 25 to 120 nm (Figure 1B), which is slightly different from the size obtained by the zeta sizer. FESEM micrographs of pristine GBA powder in both high and low resolutions are shown in Appendix A.

The FESEM samples were supplemented with FBS to see whether it would have any effect on the size and shape, since following systemic delivery, the particles can easily interact with blood serum proteins. Interestingly, the difference in morphology of GBA suspension in FBS (Figure 1) compared to that of pristine GBA (Appendix A) might be due to the dispersion of the former in water (larger space), which resulted in smaller, spherical, or irregular-shaped and less aggregated particles compared to pristine GBA. In addition, FBS might also play a role in decreasing the aggregation between particles.

### 2.4. Elemental Analysis and Functional Group Identification by Energy Dispersive X-ray (EDX) and Attenuated Total Reflection-Fourier Transform Infrared (ATR-FTIR) Spectroscopy

Elemental analysis (energy-dispersive X-ray, or EDX) of GBA suspension confirmed the presence Ca, P, C, Na, Mg, and O, which are the integral components of bone. However, a trace amount of Pt was noticed, since samples were coated with Pt for EDX analysis.

The Ca/P ratio for GBA suspension (from different sites (Figure 2)) was around 1.99 ± 0.25, which was similar or higher as compared to the known HA stoichiometric ratio at 1.67 [52]. It is plausible that some GBA particles might be in the apatite form, whereas others might change to tetracalcium phosphate (TTCP), Ca_4_O(PO_4_)_2_ at a 2.00 Ca/P molar ratio [53] (Figure 2). However, it should be noted that since GBA is regarded as a form of carbonated apatite, its Ca/P molar ratio may have the highest possible value of 3.33 [28,54].

The FTIR spectrum was used to identify different functional groups present in GBA powder. The infrared (IR) spectrum for GBA powder revealed characteristic peaks for CO_3_^2−^ at 1451 cm^−1^, 1412 cm^−1^, and 872 cm^−1^ and for PO_4_^3−^ approximately at 1000–1100 cm^−1^ and 550–650 cm^−1^ (Figure 3A) [55]. The spectrum was similar to that obtained for CA [22,42] NPs, which are also shown in Figure 3B. It also displayed peaks for OH^−^ at approximately 3261 and 3223 cm^−1^ (Figure 3A). Moreover, the characteristic peaks related to the organic portion of bone tissue were observed for amide I around 1642 cm^−1^ and amide II around 1548 cm^−1^ (Figure 3A), thus suggesting the co-existence of CA and organic traces in GBA powder. In fact, it is known that 30% of bone tissue is organic (e.g., collagen, non-collagenous proteins and lipids) and 70% is inorganic (e.g., mineralized carbonate hydroxyapatite) [56]. Thus, GBA could be regarded as an inorganic-organic hybrid material.

### 2.5. Effect of Acid Solubility on GBA Powder

To mimic the stability of GBA in tumor or tumor microenvironment, the solubility of different concentrations (0.01, 0.1, 1.00, 10, and 100 mg/mL) of GBA in both 1N HCl acid and water was analyzed, which was followed by centrifuging the samples at 13,000 rpm in a bench-top centrifuge for 20 min at 4 °C. It was observed that GBA suspensions were insoluble in water (Appendix A) but soluble in 1N HCl acid (Appendix A), as verified from the reduced size of the pellets, which is especially evident for 0.01 mg/mL and 0.1 mg/mL concentrations of GBA suspension where the GBA powder was dissolved completely in acid solution. However, for 1, 10, and 100 mg/mL concentrations of GBA suspension, the small pellet resided at the bottom of the Eppendorf tubes **(**Appendix A).

### 2.6. In Vitro pH-Dependent Dissolution Study of GBA in Bicarbonated Water

In order to exploit the pH-sensitive response of GBA, a pH-dependent dissolution study was conducted in bicarbonated water having different pHs. Instead of DMEM, bicarbonated water was used due to the tendency of GBA to be partially dissolved in DMEM. Approximately 0.1 mg/mL GBA powder was added to both DMEM and bicarbonated water, which was followed by the centrifugation of samples at 13,000 rpm for 20 min. It was observed that the size of the pellet in DMEM was smaller compared to the size of the pellet in bicarbonated water, suggesting the partial solubility of GBA powder in DMEM (Appendix A).

The pH-dependent solubility of GBA powder was further investigated through turbidity measurement at 320 nm in bicarbonated water having physiological pH (pH 7.5) and acidic pH (pH 5.5–6.5) that mimic the pH of the lysosomal/endosomal compartments. A lower turbidity of 0.05 ± 0.003 was observed for 0.1 mg/mL GBA suspension at the acidic pH of 5.5, indicating markedly faster dissolution at that acidic pH. On the other hand, GBA exhibited high turbidity (0.18 ± 0.004) at a pH of 7.5, suggesting slower dissolution (Table 3) at physiological pH. These results demonstrated the pH-sensitive nature of GBA, which can be exploited for the delivery of therapeutics.

### 2.7. pH-Dependent Release of DOX from DOX-Loaded GBA in Bicarbonated Water

To evaluate whether DOX-loaded GBA particles can effectively transport DOX to the target site and liberate DOX in a pH-responsive manner, the in vitro pH-dependent release of DOX from DOX-loaded 0.1 mg/mL GBA was investigated at different pHs. Approximately 90% of DOX was released at pH 5.5, indicating the complete disintegration of GBA at the acidic pH with 40% DOX released at pH 7.5 (Table 4). The rapid release of DOX from GBA in acidic pH can be explained by the presence of the CO_3_^2−^ and PO_4_^3−^ groups in GBA, which consumed H^+^ ions, causing a rupture in the cell membrane of the endosomes and leading to the burst effect of the endosomal compartment, thus fostering the prompt release of DOX into the cytoplasm of the cancer cell. The 40% release of DOX from DOX-loaded GBA at pH 7.5 might be due to the dissociation of the loosely bound DOX to the GBA particles following high-speed centrifugation. These results demonstrated that GBA may effectively retain the entrapped DOX until it reaches the target site of the tumor. Additionally, it may increase DOX accumulation inside cancer cells via an EPR effect, resulting in a desirable approach for targeted delivery in breast cancer cells.

### 2.8. Cell Viability Analysis by Optical Microscopy and MTT Assay

The effect of GBA on the viability of MCF-7 human adenocarcinoma cells was evaluated by 3-4, 5-dimethylthiazol-2-yl-2, 5-diphenyltetrazolium bromide (MTT) assay. Optical micrographs were used to acquire qualitative information on cell viability after 24, 48, and 72 h of treatment. Untreated cells were referred to as the control group, while the remaining groups were assessed against the control. Based on the optical micrographs, it was observed that the particles from different concentrations of GBA had been successfully internalized by the cells, which was especially evident for GBA at 0.1 mg/mL (Figure 4). In addition, the morphology of the cells looked organized, highlighting the non-toxic effect of GBA on MCF-7 cells. Higher-Resolution images for the cellular internalization of GBA at 0.1 mg/mL concentration are shown in Figure 5.

The effect of GBA (0.1 mg/mL) on MCF-7 cells after 24, 48, and 72 h of treatment is shown in higher resolution optical images as below.

The viability of GBA on MCF-7 cells was further analyzed quantitatively using MTT colorimetric assay. As illustrated in Figure 6, the cell viability of different concentrations of GBA after 24, 48, and 72 h of treatments was greater than 90%, indicating that there was no innate toxicity for all investigated concentrations. Additionally, there was no significant difference between the viability of untreated and GBA-treated cells.

### 2.9. Binding Affinity of DOX with GBA

The binding affinity of drugs to NPs plays a pivotal role in the development of an effective DDS. The drug-binding efficiency depends on the properties of both carriers and drugs, which include the molar mass and solubility of the drug, and the shape and size of the carrier, as well as the chemical interactions between the carriers and the drugs [57]. Moreover, an increase in drug-binding affinity leads to an increase in stability and cellular internalization of the drug-loaded carriers in cancer cells [58]. For example, in one study, pegylated mesoporous silica NPs showed an enhanced DOX loading efficacy of 0.98 mg (DOX)/mg (MSN) [59], which facilitated the apoptosis in cancer cells. In another study, a composite of bioengineered silk, EMS2, and magnetic iron oxide (IO) NPs displayed more than two-fold higher DOX-loading capacity compared to plain EMS2 spheres, which catalyzed their potential as a DDS [60]. In addition, pectin-capped gold nanoparticles (PEC-AuNPs) demonstrated a heightened loading efficiency of around 78% [61]. Other DDSs for treating cancer including polymers, micelles, lipid, CeO_2_, etc. also exhibited good binding affinity for DOX [58,62,63,64].

The binding affinity of DOX to GBA at two concentrations (0.01 and 0.1 mg/mL) in carbonated water (pH adjusted to 7.5) was determined over a range of DOX concentrations (2, 4, 6, 8, and 10 μM) (Figure 7). Based on Figure 7, it is clear that the binding affinity was affected by the concentration of DOX added; i.e., there is a drastic increase in the binding affinity of DOX in both GBA concentrations (0.01 and 0.1 mg/mL) with increasing concentrations of DOX. However, the binding affinity of DOX for 0.01 mg/mL GBA was significantly higher compared to that for 0.1 mg/mL GBA. This might be due to the presence of less aggregates at 0.01 mg/mL GBA, which facilitated their binding compared to higher concentration (0.1 mg/mL GBA), which revealed larger aggregates. The larger aggregates might reduce the binding of DOX with GBA by reducing the surface-to-volume ratio of the particles. For example, for DOX at 10 μM, the binding affinity of DOX with 0.01 mg/mL GBA was 23% ± 1.3 compared to only 14% ± 0.2 for 0.1 mg/mL GBA. The binding affinity of DOX to GBA-like CA NPs [21] is dependent on the chemical reaction between DOX (positively charged) and CO_3_^2−^ and PO_4_^3−^ of GBA (both are negatively charged).

### 2.10. Qualitative and Quantitative Study of Cellular Internalization of DOX-Loaded GBA in MCF-7 and MDA-MB-231 Cells

The barriers presented by the cell membrane pose a challenge for NPs to penetrate cancer cells in order to reach the sites of action [65]. For NPs-assisted cancer therapy, the effective accumulation of drug-loaded NPs in tumor is crucial. To evaluate the ability of DOX-loaded GBA to be internalized into cancer cells, qualitative DOX-loaded GBA (0.01 and 0.1 mg/mL) cellular uptake was visually confirmed in MDA-MB-231 and MCF-7 cells in a predetermined time interval of 4 h. A higher fluorescence was seen in both concentrations of DOX-loaded GBA compared to free DOX in both cell lines (Figure 8A,B). However, the fluorescence from DOX-loaded GBA was stronger and brighter in MDA-MB-231 cells (Figure 8B) as opposed to MCF-7 cells (Figure 8A), indicating that GBA can be rapidly internalized into the cells.

Having analyzed cellular uptake qualitatively, quantitative cellular uptake on GBA loaded with DOX was conducted where the determination of fluorescence of cell lysates was done. There was significantly higher DOX-loaded cellular uptake in both GBA concentrations (0.01 and 0.1 mg/mL) for both cell lines as compared to that of free DOX (Figure 9). Nevertheless, the internalizing effect of DOX loaded with 0.1 mg/mL GBA in MCF-7 cells was more compared to DOX loaded with 0.01 mg/mL GBA (Figure 9). However, relatively more fluorescence intensity was observed for DOX loaded with 0.01 and 0.1 mg/mL GBA in MDA-MB-231 cells than in MCF-7 cells. These findings are correlated with the observation made with fluorescence microscopy. Thus, we conclude that GBA promotes the cellular uptake of DOX inside the cancer cells.

### 2.11. In Vitro Cytotoxicity Analysis of DOX-Loaded GBA

To confirm whether DOX-loaded GBA can retain its tumor-killing activity, an in vitro cytotoxicity test was conducted in both types of cells after two days of treatment. In both types of cell lines, similar cell viability or toxicity was seen for DOX-loaded GBA and free DOX. The highest toxic effect was reported for 1 µM DOX concentration in both MCF-7 and MDA-MB-231 cells, although the values of DOX-loaded GBA were not different from that for free DOX (Figure 10).

The enhancement of cytotoxicity was further assessed for DOX loaded into GBA at 0.01 and 0.1 mg/mL (Table 5). The cytotoxicity enhancement seen with DOX loaded into 0.1 mg/mL GBA was slightly greater than DOX loaded into 0.01 mg/mL in MCF-7 cells. Nevertheless, the opposite effect was seen in MDA-MB-231 cells. However, at 1 μM DOX concentration, GBA exhibited the highest cell death, with cytotoxicity enhancement of 71.1 ± 6.1 (DOX-loaded 0.1 mg/mL GBA) and 81.3 ± 4.7 (DOX-loaded 0.01 mg/mL GBA) in MDA-MB-231 cells (Table 5).

The cell viability of DOX-loaded GBA particles was also assessed with 25,000 cells/well (MCF-7 cells), and the data were obtained after 48 h of treatment. The difference in toxic effect between DOX-loaded GBA and free DOX was more evident with 25,000 cells/well (Figure 11) compared to 50,000 cells/well (Figure 10). A significantly enhanced toxic effect was observed for DOX-loaded GBA particles compared to free DOX. The MCF-7 cells demonstrated mitigated cell viability on exposure to DOX-loaded GBA having 1 μM DOX concentration comparable to DOX-loaded GBA having 100 nM and 500 nM DOX concentrations, respectively. Indeed, the toxic effect of DOX-loaded GBA was concentration-dependent. The cytotoxicity enhancement data for DOX-loaded GBA particles are shown in Table 6. The maximum cytotoxicity enhancement of 58 ± 2.8 and 62 ± 1.4 was observed with 0.01 and 0.1 mg/mL GBA at 1 μM DOX concentration (Table 6).

### 2.12. Protein Corona Analysis

Once the NPs are administered inside the body, they interact with the blood circulatory system, which causes the blood serum proteins to coat the surface of the NPs. The resultant protein corona affects the biodistribution as well as the half-life of the NPs, thereby influencing their tumor targetability and therapeutic efficacy [28,66,67]. Hence, quantitative protein corona analysis helps us understand how NPs behave biologically. In this study, the profiling of protein corona for different concentrations (0.01, 0.1, 1 mg/mL) of GBA in 10% FBS was investigated by an in-solution digestion, followed by Q-TOF LC-MS/MS. The detected serum proteins, their functions, as well as their isoelectric point (pI) are listed in Table 7. The pie chart summarizes protein taxonomy (stated as a percentage of biological functions) (Figure 12).

Based on Table 7 and Figure 12, GBA at a high concentration (1 mg/mL) has a high affinity for transport proteins but no affinity for structural proteins. On the other hand, at a low concentration (0.01 mg/mL), it possesses high affinity toward structural proteins but less affinity toward transport proteins. Interestingly, 0.1 mg/mL GBA exhibited good affinity only toward the hemoglobin alpha A subunit. Although both concentrations (0.1 and 1 mg/mL) exhibited affinity toward a few other selective proteins, high affinity toward transport proteins might increase the drug circulation time in the blood, resulting in enhanced tumor accumulation via the EPR effect and an increased internalization of GBA particles into breast cancer cells. Interestingly, GBA showed no interaction with opsonins (i.e., fibrinogen, complement factors, and IgG), which could otherwise facilitate the cellular uptake of GBA particles through phagocytosis due to the presence of opsonin-cognate receptors expressed on the surface of the phagocyte [68,69], resulting in the clearance of GBA particles from systemic circulation [70] and fostering off-target distribution in major reticuloendothelial system organs (e.g., the liver, lungs, and spleen) [71].

When the pI value is less than 7.5, corona proteins are negatively charged and will be acidic, thus allowing electrostatic interaction with GBA. On the other hand, corona proteins with pI > 7.5 were positively charged and basic in physiological pH, thus increasing their chance of electrostatically adhering to the anionic domains (e.g., OH^−^, CO_3_^2−^, and PO_4_^3−^) of GBA.

## 3. Materials and Methods

### 3.1. Materials

Sodium bicarbonate (NaHCO_3_) (purity: 99.5–100.5%), doxorubicin hydrochloride (DOX) (purity: 98–102%), Dulbecco’s modified eagle medium (DMEM), dimethyl sulfoxide (DMSO) (≥99.9% purity), ethylenediaminetetraacetic acid (EDTA) (99% purity), and thiazolyl blue tetrazolium bromide (MTT) (97% purity) were bought from Sigma-Aldrich (St. Louis, MO, USA). DMEM powder, trypsin-ethylenediaminetetraacetate (trypsin-EDTA), fetal bovine serum (FBS) (purity: certified) and penicillin-streptomycin were purchased from Gibco by Life Technology (Thermo Fischer Scientific, Waltham, MA, USA). Hydrochloric acid (HCl) (purity: 37%) and ethanol (purity: 99.5%) were procured from Fischer Scientific (Loughborough, Leicestershire, UK). MDA-MB-231 (M.D. Anderson Metastasis Breast Cancer-231) and MCF-7 (Michigan Cancer Foundation-7) cells was bought from ATCC (Manassas, VA, USA). Goose bone ash (GBA) capsules were supplied by Star Goose Enterprise, Seri Kembangan, Selangor, Malaysia, who holds a patent (PI2017701257).

### 3.2. Preparation of GBA Suspension and Verification of Particle Aggregation through Optical Microscopic Images and Turbidity Analysis

Each capsule of GBA contained 500 mg of ash, which was retrieved and mixed with water to make a 5 mL suspension (100 mg/mL) of primary stock. The serial dilution method was used to prepare stocks of GBA suspension in the respective concentrations subsequently (10, 1, 0.1, and 0.01 mg/mL).

To prepare and optimize GBA particles, different concentrations of GBA suspension were analyzed through optical micrographs and turbidity measurements. The bicarbonated Dulbecco’s Modified Eagle’s Medium (DMEM) media was prepared by mixing sodium bicarbonate (NaHCO_3_) (44 mM) and DMEM powder (0.675 g) in Milli-Q water and the pH was adjusted to 7.5 utilizing hydrochloric acid (0.1 M). Different concentrations of GBA (0.01 mg/mL, 0.10 mg/mL, 0.01 µg/mL, 0.10 µg/mL, and 1.00 µg/mL) were prepared in DMEM (1 mL) and subsequently incubated at 37 °C for 30 min. Then, the suspensions were transferred to 24-well plates. Then, optical micrographs were captured by using an Olympus IX81 fluorescence microscope (Shinjuku, Tokyo, Japan). The magnification used was 10× (scale bar of 50 µm). The turbidity analysis was conducted at different GBA concentrations (0.01, 0.10, 1.00, 10.00, and 100.00 mg/mL) in water by employing 1800 MS UV-VIS spectrophotometer (Shimadzu, Japan) at 320 nm. Data were expressed as mean ± standard deviation for the triplicates.

### 3.3. Particle Size Measurement of GBA Suspension by Dynamic Light Scattering (DLS) Method

A Malvern nano zeta sizer (Malvern, Worcestershire, UK) was used to measure the size by dynamic light scattering (DLS) for different concentrations (1 μg/mL, 0.01 mg/mL, 0.10 mg/mL, 1.00 mg/mL, 10.00 mg/mL, and 100.00 mg/mL) of GBA suspension in water after a 30 min incubation step (37 °C) followed by the addition of FBS (10%). A zeta sizer software was used to analyze the data, which was shown as mean ± standard deviation for the duplicate samples.

### 3.4. Field Emission Scanning Electron Microscopic (FESEM) Analysis

Approximately 2 µL each of GBA suspension in water (1 and 10 mg/mL) was dried at room temperature for 40 min. Subsequently, it was positioned on a sample holder and layered with an adhesive tape of carbon, followed by coating with platinum (Pt) sputtering with a 30 mA sputter current for 40s at 2.30 tooling factor. Then, the shape, size, and morphological characteristics of the respective GBA suspension were captured by FESEM (2 kV) (Hitachi/SU8010, Tokyo, Japan). FESEM pictures for pristine GBA powder were similarly captured.

### 3.5. Energy-Dispersive X-ray (EDX) and Attenuated Total Reflection-Fourier Transform Infrared (ATR-FTIR) Spectroscopic Analysis

GBA suspension (10 mg/mL) was investigated using EDX (X-max, 50 mm, Horiba, Japan) suspended in water and received a shaft of energy at 15 kV (input count rate: 100,000) in order to determine the composition and the elements present. For this analysis, the highest concentration (10 mg/mL) was used due to its better overall sensitivity and minimal detection limit for EDX (which was approximately 0.1 wt % (for all elements). The data generated represented the spectra related to the peaks of the different elements present in GBA samples.

Attenuated total reflection-Fourier transform infrared (ATR-FTIR) spectroscopic analysis of GBA powder was executed using a Varian 640-IR with a spectral window of 4000–400 cm^−1^ in transmission mode. Data were analyzed through Varian Resolution Pro 640 software (Agilent, Santa Clara, CA, USA).

### 3.6. pH-Responsive Dissolution Study of GBA in Water and 1N HCl

Different concentrations (0.01, 0.1, 1, 10, and 100 mg/mL) of GBA powder in both water and 1N HCl were prepared, followed by centrifuging the samples at 13,000 rpm in a bench-top centrifuge for 20 min at 4 °C. Then, the solubility of the respective samples was determined.

### 3.7. pH Sensitivity of GBA in DMEM and Bicarbonated Water

GBA (0.1 mg/mL) was prepared in DMEM and bicarbonated water. The solution was centrifuged at 13,000 rpm for 20 min. Subsequently, the size of the pellets was observed to determine the effect of GBA suspension in DMEM and bicarbonated water.

For further analysis, GBA (0.1 mg/mL) was prepared in bicarbonated water. Approximately 0.1 mg GBA powder was added to freshly prepared bicarbonated water (200 µL), prior to half an hour incubation at 37 °C. Subsequently, bicarbonated water (800 µL) was added at different pHs (7.5, 7, 6.5, 6.0, and 5.5). Then, the turbidity was quantified via an 1800 MS UV-VIS spectrophotometer (Shimadzu, Japan) at 320 nm. Samples were prepared in triplicate before the estimation of turbidity as mean ± standard deviation.

### 3.8. pH-Dependent Release of DOX from DOX-Loaded GBA

DOX-Loaded GBA was prepared by adding 5 µM DOX to 0.1 mg/mL GBA in 200 µL freshly prepared bicarbonated water, followed by a 30 min incubation step (37 °C). Subsequently, bicarbonated water (800 µL) at several pHs (5.5, 6.0, 6.5, 7.0, and 7.5) was topped up. The sample was subjected to cold (4 °C) centrifugation (12,000 rpm × 20 min) in a micro-centrifuge. Excitation/emission (485/535 nm) was measured by a 2030 multilabel reader (×5) that was connected to a PerkinElmer 2030 software (PerkinElmer, Waltham, MA, USA). The amount of DOX released from DOX-loaded GBA at different pHs was calculated using the standard curve and the given formula:(1)$$\mathrm{DOX}\;\mathrm{Released}\;\mathrm{from}\;\mathrm{GBA}\;\left(\%\right)=\frac{B}{A}\times 100\%$$
where

A = DOX released from DOX-loaded GBA at pH 7.5

B = DOX released from DOX-loaded GBA at different pHs (7.5, 7.0, 6.5, 6.0, and 5.5)

The experiment was repeated thrice, and the results were analyzed in terms of mean value and standard deviation.

### 3.9. Cell Culture and Seeding

The human mammary cell line MCF-7 was cultured in flasks (25 cm^2^) in complete DMEM (CDMEM) media, which has FBS (10%), penicillin (1%), as well as an antibiotic named streptomycin at pH 7.4. The mixture was subsequently incubated in a humidified atmosphere at 37 °C with 5% carbon dioxide (CO_2_). The MCF-7 cells were trypsinized, washed, centrifuged, and seeded in 24-well plates (Greiner, Frickenhause, Germany) containing 50,000 cells/well followed by an overnight incubation at 37 °C containing 5% CO_2_.

### 3.10. Cell Treatment with GBA Suspension

Different concentrations (0.1 mg/mL, 0.01 mg/mL, 1 µg/mL, 0.1 µg/mL, and 0.01 µg/mL) of GBA suspension were prepared in freshly made bicarbonate DMEM solution (1 mL, buffered at pH 7.4) that was filtered. Following that, the suspensions were incubated for 30 min (37 °C), followed by supplementation with 10% FBS. Then, the DMEM from each well was replaced with GBA suspension prepared in bicarbonate buffered CDMEM, followed by incubation for 24, 48, and 72 h at 37 °C in a humidified atmosphere (5% CO_2_).

### 3.11. Cell Viability and Cytotoxicity Assessment by MTT (3-4, 5-Dimethylthiazol-2-yl-2, 5-Diphenyltetrazolium Bromide) Assay

MTT assay was used to evaluate the biocompatibility of GBA. Approximately 50 μL of MTT [5 mg/mL in phosphate-buffered saline (PBS)] was supplemented in each well in the 24-well plate treated with GBA to develop purple formazan crystals, followed by incubation at 37 °C for 4 h. The media containing MTT solution was discarded from each well. Then, DMSO solution (300 μL) was added to solubilize the purple crystals. Subsequently, sample absorbances were measured using a spectrophotometric microplate reader (BIO-RAD-Microplate Reader, Hercules, CA, USA) at an optical wavelength and a reference wavelength of 595 and 630 nm, respectively. Each sample was prepared in triplicate. The results were given as mean ± standard deviation. The given formula was used to calculate cell viability (%):(2)$$\mathrm{Cell}\;\mathrm{Viability}\;\left(\%\right)=\frac{{\mathrm{Absorbance}}_{\mathrm{GBA}-\mathrm{bound}\;\mathrm{drug}}-{\mathrm{Absorbance}}_{\mathrm{free}\;\mathrm{drug}}}{{\mathrm{Absorbance}}_{\mathrm{GBA}}-{\mathrm{Absorbance}}_{\mathrm{free}\;\mathrm{drug}}}\;\times \;100.$$

For cytotoxicity analysis, MCF-7 cells (50,000 cells per well), MDA-MB-231 cells (50,000 cells per well), and MCF-7 cells (25,000 cells per well) were seeded in 24-well plates. Following incubation of cells at 37 °C in 5% CO_2_ for 24 h, the growth medium was removed, and the cells were treated with DOX-loaded 0.01 mg/mL GBA and DOX-loaded 0.1 mg/mL GBA, which were prepared using different DOX concentrations (100 nM, 500 nM, or 1 µM). Controls include different concentrations (100 nM, 500 nM, or 1 µM) of free DOX and DOX-free medium.

After 48 h, the cells were incubated with 50 µL of MTT solution for 4 h. Then, 300 µL of DMSO was added to each well to solubilize the formazan crystals. Then, the absorbance of each well was measured at an optical/reference wavelength of 595/630 nm using a spectrophotometric microplate reader (BIO-RAD-Microplate Reader, Hercules, CA, USA). Each of the samples was prepared in triplicate. The results were given as mean ± standard deviation. The below given formula was used to calculate cell viability (%):(3)$$\mathrm{Cell}\;\mathrm{Viability}\;\left(\%\right)=\frac{{\mathrm{Absorbance}}_{\mathrm{GBA}\;\mathrm{bound}\;\mathrm{drug}}-{\mathrm{Absorbance}}_{\mathrm{free}\;\mathrm{drug}}}{{\mathrm{Absorbance}}_{\mathrm{GBA}}-{\mathrm{Absorbance}}_{\mathrm{free}\;\mathrm{drug}}}\times 100.$$

### 3.12. Binding Affinity of DOX to GBA

The binding affinity of DOX to GBA was evaluated by adding different concentrations (2, 4, 6, 8, and 10 µM) of DOX to 0.01 and 0.1 mg/mL GBA suspension in bicarbonated water (pH adjusted to 7.5), followed by incubation at 37 °C for 30 min. Then, the mixtures were centrifuged at 12,000 rpm for 20 min at 4 °C by using a bench-top micro-centrifuge. Approximately 200 µL of supernatant from individual samples were subsequently kept in a 96 well-plate. Subsequently, the samples were exposed to a 2030 multilabel plate reader and were assessed at 485/535 nm as previously before. The concentration of unbound DOX was established using a standard curve. The percentage of binding of DOX with GBA was evaluated based on the following formula:(4)$$\mathrm{DOX}\;\mathrm{Binding}\;\left(\%\right)=\frac{A-B}{A}\times 100$$
where

A = total concentrations of DOX used in the experiment (i.e., 2, 4, 6, 8, and 10 µM)

B = concentration of unbound DOX in the GBA suspension

The experiment was conducted in triplicate, and the results were evaluated as mean ± standard deviation.

### 3.13. Visualization of Cellular Uptake of DOX-Loaded GBA Using Fluorescence Microscopy

The seeding of MCF-7 and MDA-MB-231 cells (50,000 cells/well) was done in 24-well plates, prior to incubation for 24 h at 37 °C in 5% CO_2_. Then, the wells were treated with 5 µM free DOX, DOX (5 µM)-loaded 0.01 mg/mL, and 0.1 mg/mL GBA suspension in DMEM (pH 7.4), which were incubated at 37 °C for half an hour and supplemented with 10% FBS. Then, the CDMEM from the seeded 24-well plates was replaced by DOX-loaded GBA suspension. After 4 h, the supernatant of the culture media was removed, and the cells were washed with 100 µL of EDTA (10 mM) in PBS to eliminate any extracellular particles, followed by washing twice using 100 µL PBS. Then, the images of the cells were captured in the presence of 100 µL PBS using the Olympus Fluorescence microscope IX81 (Shinjuku, Tokyo, Japan) attached to a CellSens Dimension software (Waltham, MA, USA).

### 3.14. Quantitative Analysis of Cellular Uptake for DOX-Loaded GBA

For quantitative analysis, the PBS from each well was discarded, and the cells were lysed with 100 µL lysis buffer. Then, the fluorescence intensity of the lysate was measured at an excitation/emission wavelength of 485/535 nm using a 2030 multi-labeled reader victor X5 (PerkinElmer, Waltham, MA, USA) attached with the PerkinElmer 2030 manager software. Then, the cellular uptake of drugs internalized by the cells was calculated utilizing the standard curve and the given formula:(5)$$\mathrm{Cellular}\;\mathrm{Uptake}\;\left(\%\right)=\frac{\mathrm{Fluorescence}\;\mathrm{intensity}\;\mathrm{of}\;\mathrm{DOX}-\mathrm{loaded}\;\mathrm{GBA}\;\mathrm{internalized}\;\mathrm{into}\;\mathrm{the}\;\mathrm{cells}}{\mathrm{Fluorescence}\;\mathrm{intensity}\;\mathrm{of}\;\mathrm{free}\;\mathrm{DOX}}\times 100.$$

The experiment was conducted in triplicate, and the results were evaluated in terms of mean value and standard deviation.

### 3.15. Protein Corona Analysis

#### 3.15.1. Protein Corona Analysis of GBA through In-Solution Digestion

To 0.01, 0.10, and 1.00 mg/mL of GBA suspension each in water, 10% FBS was added, and the mixtures were incubated for 20 min at 37 °C, followed by centrifuging at 13,000 rpm for 15 min, discarding the supernatants, and washing the pellets twice in Milli Q water. Then, the pellets were dissolved in 100 µL of 50 mM EDTA in water. Following this, 25 µL of 100 mM (NH4)_2_CO_3_ solution and tetrafluoroethylene (TFE) were added to the protein mixture with 1 µL of 200 mM dithiothreitol (DTT) solution. Then, the mixtures were vortexed and heated for 60 min at 60 °C. Subsequently, the mixtures were supplemented with 4 µL of 200 mM iodoacetamide (IAM) and 100 µL of 100 mM (NH4)_2_CO_3_ solution followed by an incubation in the dark for 60 min. Then, MS Grade 25 µL of trypsin (1 µg/mL) was added to allow for digestion for 18 h. Subsequently, the mixture was supplemented with 1 µL of formic acid and speed-vacuumed for 18 h, followed by Q-TOF LC-MS/MS analysis.

#### 3.15.2. Sample Preparation for Mass Spectrometry-Based Proteomics

To the dry peptide digest, 0.1% formic acid (10 µL) was added, followed by sonication for 10 min and centrifugation at 14,000× *g* for 5–10 min. Finally, 5 µL supernatant in an MS tube was subjected to liquid chromatography quadrupole time-of-flight mass spectrometry (LC-QTOF) auto-sampler for further analysis.

#### 3.15.3. High-Efficiency Nano-Flow Liquid Chromatography Electrospray-Ionization Coupled with Mass Spectrometry

An agilent Poroshell 300Å pore C18 column (Agilent, Santa Clara, CA, USA) was used to load the peptide digest in the presence of formic acid (0.1%). Gradients of 5% solution B (90% acetonitrile in 0.1% formic acid) over 0–30 min and 75% solution B over 30–39 min were used to elute the peptides. The fragmented voltage of 1750 V and 360 V was maintained with 5 L/min of gas flow temperature of 325 °C with a positive quadrupole-time of flight (Q-TOF). Auto MS mode was fixed at between 110 and 3000 *m*/*z* (MS) or 50 and 3000 *m*/*z* (MS/MS) to measure peptide spectrums with acquisition values of two spectra/s (MS) or four spectra/s (MS/MS). Data were obtained from an Agilent MassHunter (Agilent Technologies, Santa Clara, CA, USA) coupled with PEAKS 8.0 software (Bioinformatics Solutions Inc., Waterloo, ON, Canada) for spectrum determination.

#### 3.15.4. Protein Recognition and Quantification

A SwissProt.Mus_musculus database search was conducted for protein identification. Subsequently, homology was explored using PEAKS Studio 8.0 (Bioinformatics Solution Inc., Waterloo, ON, Canada). The maximum mixed cleavages of three were utilized for carbamidomethylation with an error tolerance of 0.1 Da for both parent mass and fragment mass. Digestion was done using trypsin. Filtration for inaccurate proteins was done by a false discovery rate (FDR) of 1% and having unique peptides ≥1. A value of −10LogP > 20 showed that detected proteins were high in confidence.

PEAKS Q protein quantification software was used to quantify the relative differential changes of proteins in DBA by a quantification technique that is label-free with reference to the recognized comparative intensities of peptide ion peaks. The expectation maximization (EM) algorithm was applied to detect the feature on each sample separately. The identical peptide features of separate samples were evaluated through a retention time alignment algorithm that was high performance. In the heat map summary, the groups were color-coded to identify the clusters between the four NPs and the intensity of any peptide that is quantifiable. The −10LogP value signifies the importance of peptide. A *p*-value of 0.01 and a cut-off value of 20 were regarded as significant. A two-fold change by threshold ratio was demonstrated by 1. The heat map showed filtered proteins for quantitative analysis. The relative protein abundance of each protein was depicted as characteristic protein heat maps. Aggregation was observed if the representative proteins have similar expressions. Hierarchical clustering was exploited using a neighbor-joining algorithm at similar Euclidean distance to the log 2 ratios of abundance with respect to the mean abundance.

### 3.16. Statistical Analysis

The findings were stated as mean ± standard deviation. Statistical analysis was conducted by using GraphPad software (San Diego, CA, USA) through an independent t-test to assess the *p* values at 95% confidence interval between treated (GBA) and untreated group groups (for cell viability analysis) and between DOX-loaded GBA and the control (free DOX) (for cellular uptake analysis).

## 4. Conclusions

Our findings demonstrated that GBA can enhance cellular internalization and drug-binding efficacy when used as a carrier. GBA particles also have no innate toxicity. In vitro cytotoxicity analysis indicated that DOX-loaded GBA particles can successfully inhibit the proliferation of breast cancer cells (e.g., MCF-7 and MDA-MB-231 cells). The pH-responsive release study of GBA confirmed its pH-responsive nature by releasing DOX in acidic pH (endosomal/lysosomal pH: 6.5–5.5), while remaining stable in physiological pH (pH 7.5). The protein corona analysis indicated a higher affinity toward transport proteins, structural proteins, and a few other selective proteins. The adsorption of transport proteins on the surface of the NPs might increase the half-life of drug-loaded NPs in blood and thereby augment their accumulation in the tumor through the EPR effect, indicating that GBA can be a good potential carrier for the delivery of anticancer drugs in breast cancer cells with no/minimal side effects. Nevertheless, an in vivo study in cancer models should be carried out in the future to evaluate its potential as a DDS.

## Figures and Tables

**Figure 1 ijms-21-06721-f001:**
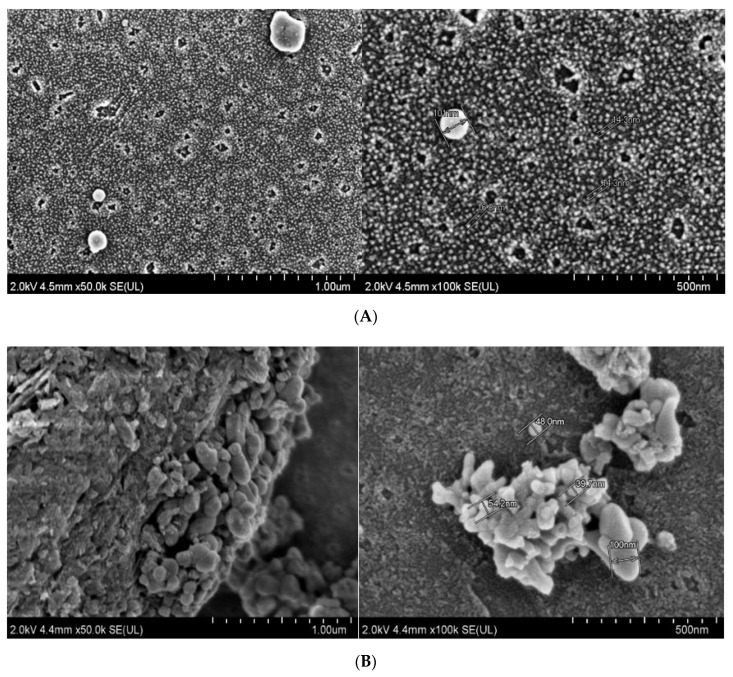
FESEM micrographs of (**A**) GBA suspension in 1 mg/mL and (**B**) GBA suspension in 10 mg/mL. The suspension was prepared in water, followed by incubation at 37 °C for 30 min and supplementation with 10% FBS. The FESEM micrographs of GBA suspension were captured at both low resolution (1 µm) and high resolution (500 nm).

**Figure 2 ijms-21-06721-f002:**
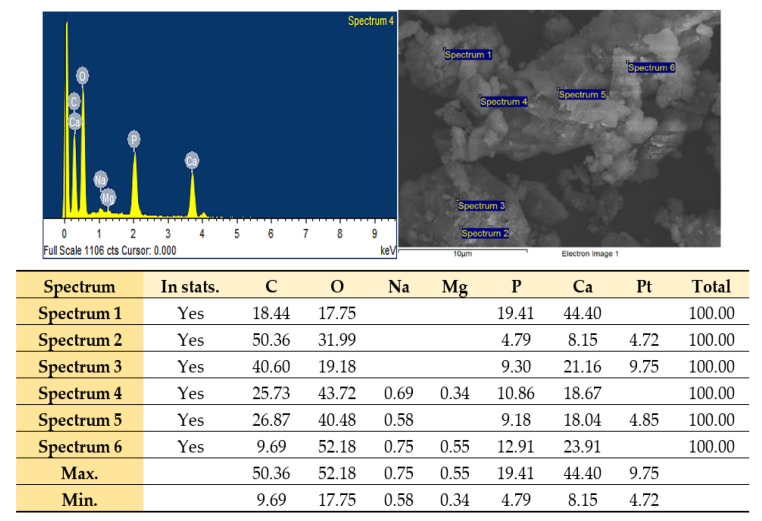
Representative EDX spectra for elemental analysis of GBA suspension.

**Figure 3 ijms-21-06721-f003:**
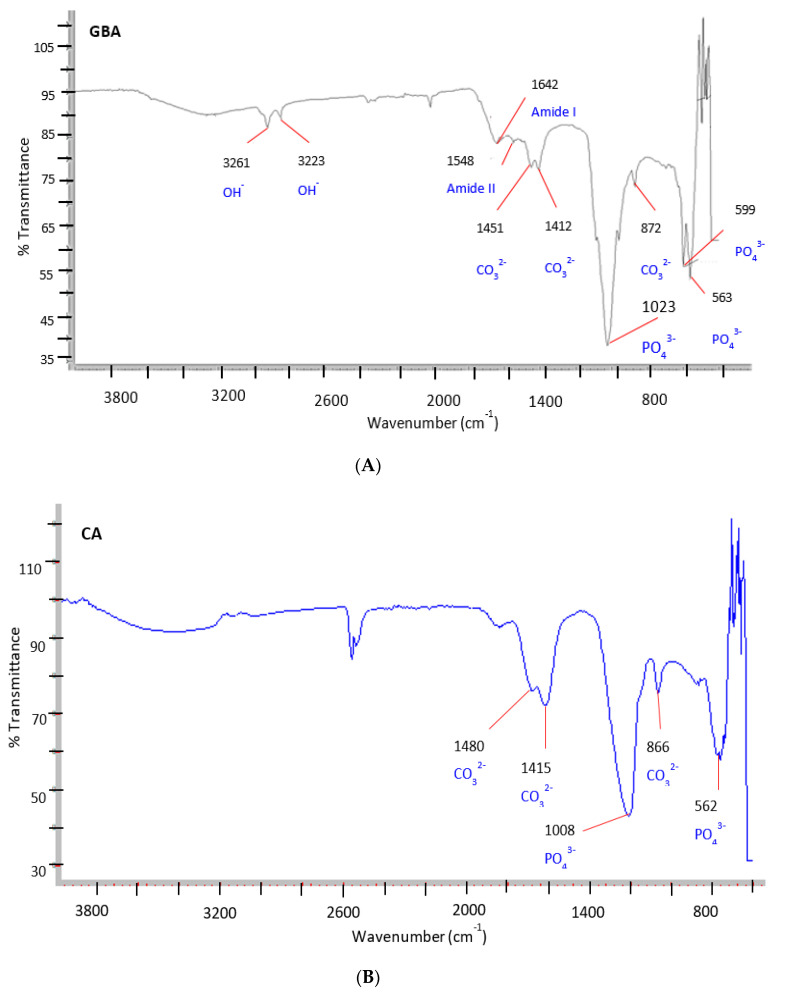
Main attenuated total reflection-Fourier transform infrared (ATR-FTIR) bands for the functional groups in (**A**) pristine GBA powder (**B**) carbonate apatite (CA).

**Figure 4 ijms-21-06721-f004:**
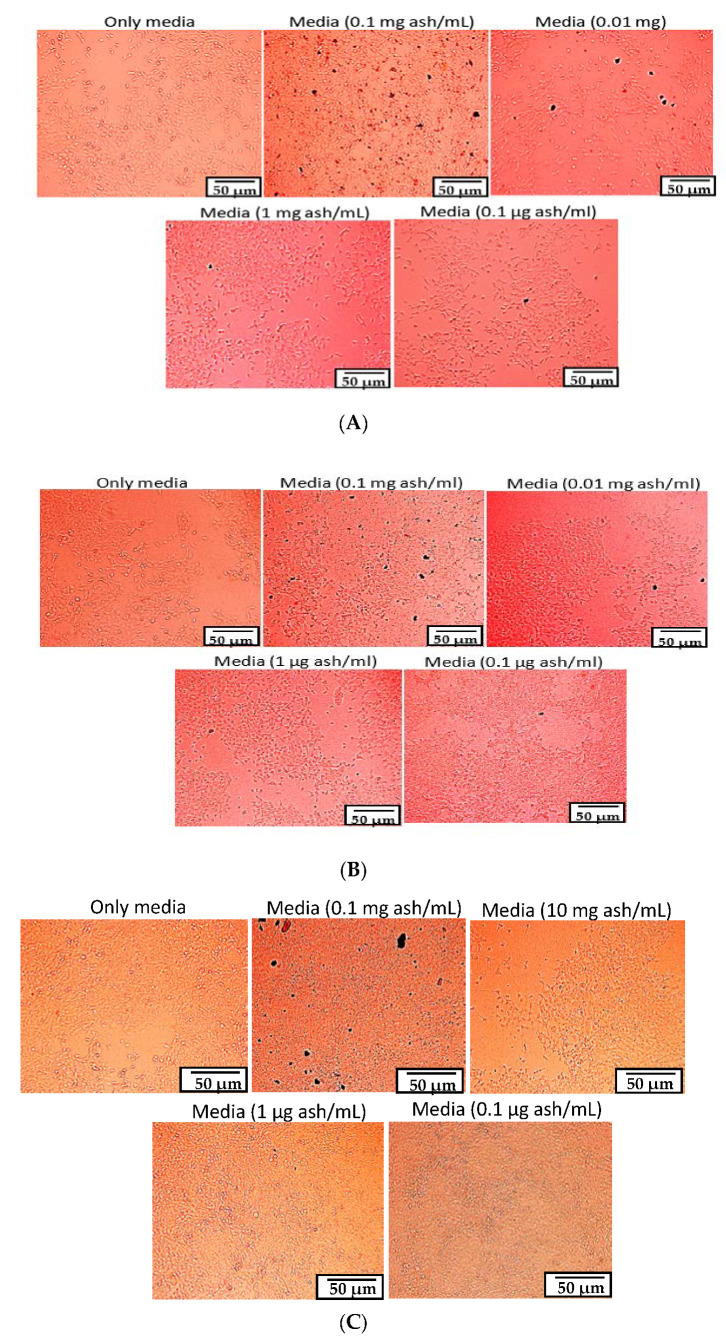
Representative images of cell viability on MCF-7 cells after (**A**) 24 (**B**) 48 and (**C**) 72 h of treatments with different concentrations of GBA. Scale bar denotes 50 µm.

**Figure 5 ijms-21-06721-f005:**

Representative images in higher resolution showing the effect of GBA (100 µg/mL) on MCF7 cells after (**A**) 24 h (**B**) 48 h, and (**C**) 72 h of treatment. Scale bar: 20 μm.

**Figure 6 ijms-21-06721-f006:**
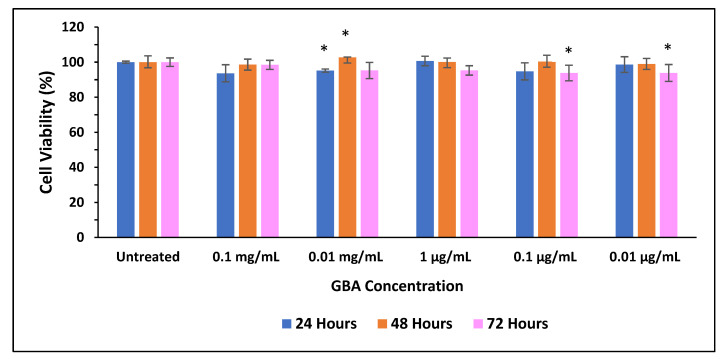
Effect of GBA on MCF-7 cell viability. Approximately 50,000 cells were seeded per well, and after 24 h, they were treated with GBA in DMEM. 3-4, 5-Dimethylthiazol-2-yl-2, 5-diphenyltetrazolium bromide (MTT) assay was performed after 24, 48, and 72 h. Cell viability was measured as compared to untreated cells. *p* values between 0.01 and 0.05 (*) were represented to be significant.

**Figure 7 ijms-21-06721-f007:**
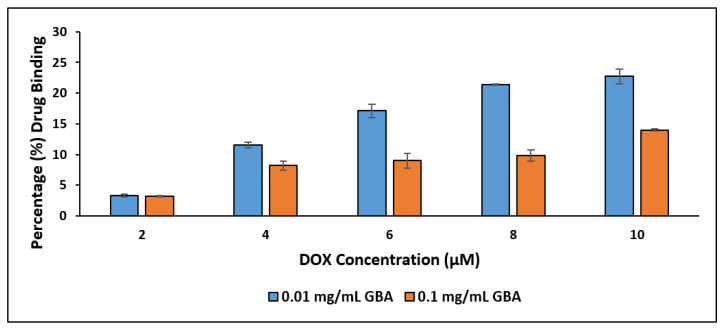
Binding affinity of 0.01 mg/mL and 0.1 mg/mL GBA with different concentrations (2, 4, 6, 8, and 10 μM) of DOX.

**Figure 8 ijms-21-06721-f008:**
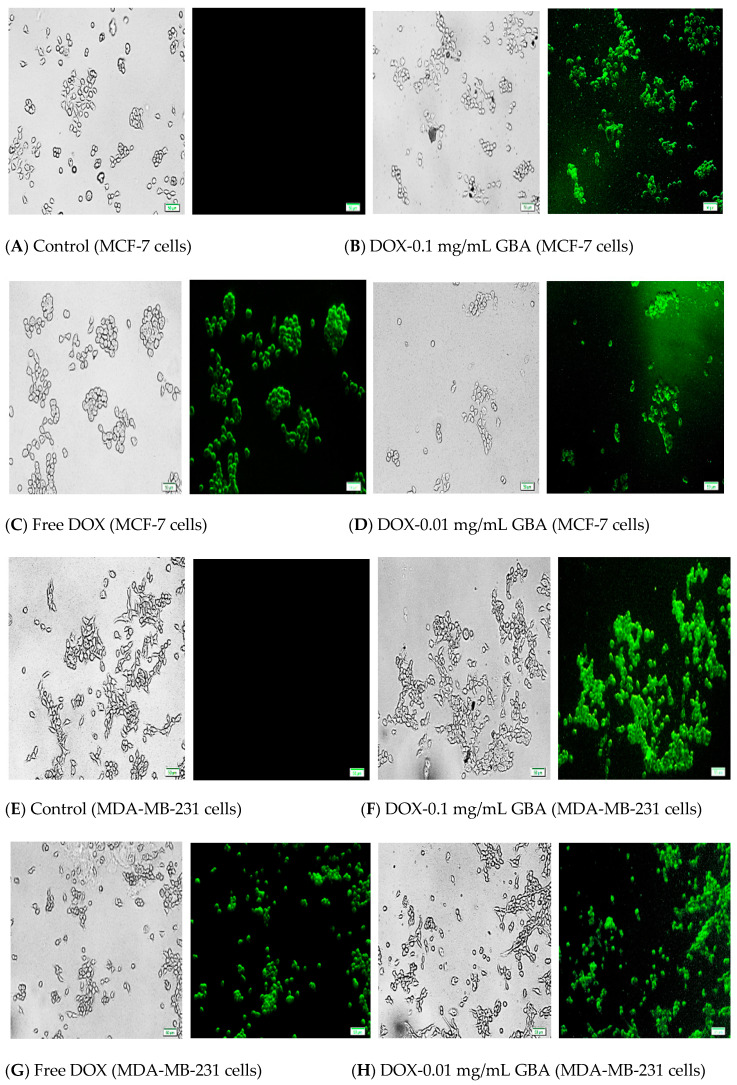
Representative fluorescent images of cellular uptake for (**A**) Control (MCF-7 cells) (**B**) DOX-0.1 mg/mL GBA (MCF-7 cells) (**C**) Free DOX (MCF-7 cells) (**D**) DOX-0.01 mg/mL GBA (MCF-7 cells) (**E**) Control (MDA-MB-231 cells) (**F**) DOX-0.1 mg/mL GBA (MDA-MB-231 cells) (**G**) Free DOX (MDA-MB-231 cells) (**H**) DOX-0.01 mg/mL GBA (MDA-MB-231 cells) following 4 h of treatment.

**Figure 9 ijms-21-06721-f009:**
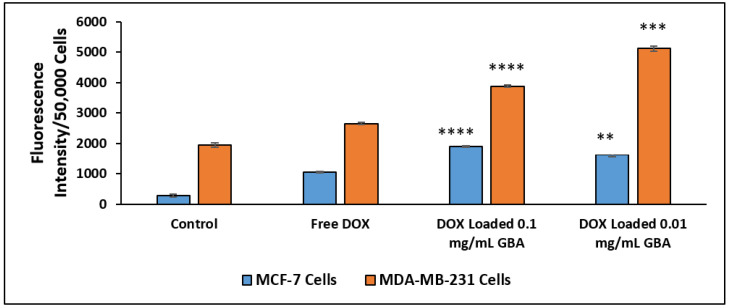
Fluorescent intensity measurement of free DOX, DOX loaded with 0.01 and 0.1 mg/mL GBA after 4 h of treatment in MCF-7 and MDA-MB-231 cells. *p* values between 0.001 and 0.01 (**) were represented to be very significant, *p* values between 0.0001 and 0.001 (***) were represented to be highly significant, and *p* values ˂ 0.0001 (****) were represented to be extremely significant.

**Figure 10 ijms-21-06721-f010:**
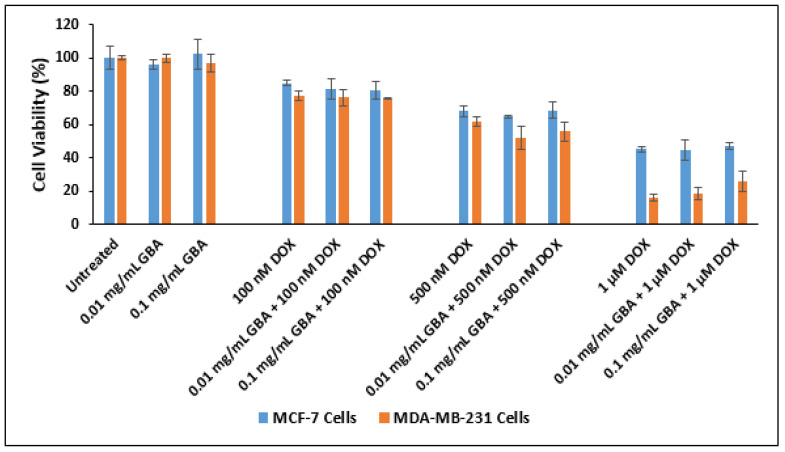
Effect of DOX-loaded GBA on cell viability based on MTT assay after 48 h of treatment in MCF-7 and MDA-MB-231 cells. Approximately 50,000 cells were seeded per well in 24 well plates. Cells were treated after 24 h with (DOX) doxorubicin (100 nM, 500 nM or 1 µM) in free form and in complexation with GBA at 0.01 mg/mL or 0.1 mg/mL. Cell viability was calculated using the absorbance of untreated cells as 100%.

**Figure 11 ijms-21-06721-f011:**
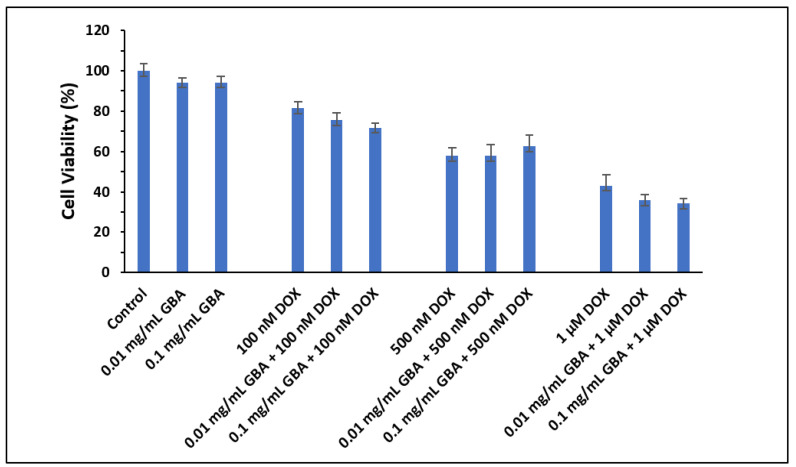
Effect of DOX-loaded GBA on the viability of MCF-7 cells seeded at 25,000 cells/well after 48 h of treatment. Approximately 25,000 MCF-7 cells were seeded per well in 24-well plates. Cells were treated after 24 h with (DOX) doxorubicin (100 nM, 500 nM, or 1 µM) in free form and in complexation with GBA at 0.01 mg/mL or 0.1 mg/mL. Cell viability was calculated using the absorbance of untreated cells as 100%.

**Figure 12 ijms-21-06721-f012:**
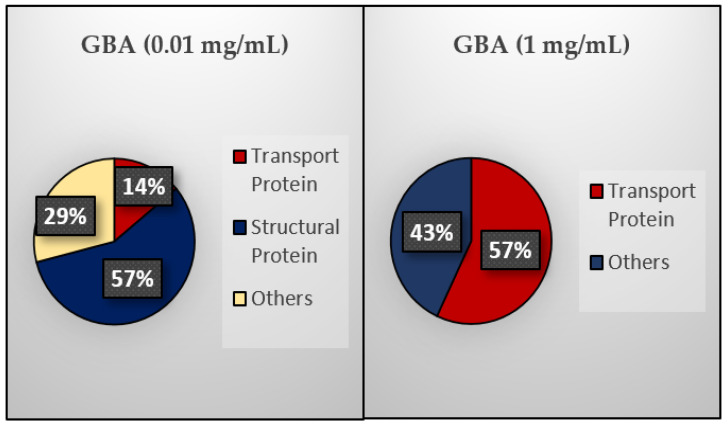
Taxonomy (%) of proteins based on their biological functions.

**Table 1 ijms-21-06721-t001:** Turbidity measurement at 320 nm for increasing concentrations (0.01, 0.1, 1, 10, and 100 mg/mL) of goose bone ash (GBA) in water after incubation at 37 °C for 30 min and supplementation with 10% fetal bovine serum (FBS).

Concentrations (mg/mL) of GBA	Turbidity at 320 nm
0.01 mg/mL	0.27 ± 0.08
0.1 mg/mL	0.73 ± 0.07
1 mg/mL	2.07 ± 0.10
10 mg/mL	3.24 ± 0.11
100 mg/mL	3.72 ± 0.13

**Table 2 ijms-21-06721-t002:** Z-Average diameter of different concentrations (1 μg/mL, 0.01 mg/mL, 0.10 mg/mL, 1.00 mg/mL, 10.00 mg/mL, and 100.00 mg/mL) of GBA suspension in water after incubation at 37 °C for 30 min and supplementation with 10% FBS.

Concentrations of GBA	Zeta Average Diameter (nm)
1 μg/mL	215 ± 18
0.01 mg/mL	477 ± 12
0.1 mg/mL	520 ± 6
1 mg/mL	946 ± 19
10 mg/mL	1021 ± 11
100 mg/mL	2373 ± 26

**Table 3 ijms-21-06721-t003:** pH-sensitive response of GBA in bicarbonated water via turbidity analysis at 320 nm.

Different pHs	pH Sensitivity (Turbidity Measurement at 320 nm)
7.5	0.18 ± 0.004
7	0.12 ± 0.009
6.5	0.08 ± 0.006
6	0.06 ± 0.004
5.5	0.05 ± 0.003

**Table 4 ijms-21-06721-t004:** pH-dependent release of doxorubicin (DOX) from GBA suspension (0.1 mg/mL) in bicarbonated water.

Different pHs	DOX Release (%)
7.5	40 ± 1.6
7	44 ± 2.2
6.5	76 ± 1.3
6	82 ± 1.6
5.5	90 ± 1.5

**Table 5 ijms-21-06721-t005:** Cytotoxicity enhancement of DOX-loaded GBA in MCF-7 and MDA-MB-231 cells.

GBA Containing Different Concentrations of DOX	Cytotoxicity Enhancement in MCF-7	Cytotoxicity Enhancement in MDA-MB-231
GBA (0.1 mg/mL) + DOX (100 nM)	21.8 ± 1.8	21.3 ± 0.1
GBA (0.01 mg/mL) + DOX (100 nM)	14.5 ± 1.3	23.5 ± 1.1
GBA (0.1 mg/mL) + DOX (500 nM)	33.8 ± 2.8	41.1 ± 3.6
GBA (0.01 mg/mL) + DOX (500 nM)	31.3 ± 0.4	48.1 ± 4.5
GBA (0.1 mg/mL) + DOX (1 µM)	55.3 ± 2.2	71.1 ± 6.1
GBA (0.01 mg/mL) + DOX (1 µM)	40.6 ± 6.6	81.3 ± 4.7

**Table 6 ijms-21-06721-t006:** Cytotoxicity enhancement of DOX-loaded GBA in MCF-7 cells (25.000/well).

GBA Containing Different Concentrations of DOX	Cytotoxicity Enhancement
GBA (0.01 mg/mL) + DOX (100 nM)	18 ± 3.5
GBA (0.1 mg/mL) + DOX (100 nM)	22 ± 0.7
GBA (0.01 mg/mL) + DOX (500 nM)	36 ± 0.7
GBA (0.1 mg/mL) + DOX (500 nM)	31 ± 6.8
GBA (0.01 mg/mL) + DOX (1 μM)	58 ± 2.8
GBA (0.1 mg/mL) + DOX (1 μM)	62 ± 1.4

**Table 7 ijms-21-06721-t007:** List of identified proteins with different concentrations (0.01, 0.1, and 1 mg/mL) of GBA in 10% FBS. (Note: √ denotes the proteins identified at various concentrations of GBA).

Protein Classes	GBA (mg/mL)			Identified Protein	pI	Detailed Function
	1	0.1	0.01			
Transport Protein	√	√	√	Hemoglobin Alpha A Subunit	8.54	Heme, iron, oxygen binding; Oxygen carrier
Transport Protein	√			Hemoglobin Beta Subunit	6.25	Heme and oxygen binding; Oxygen carrier
Transport Protein	√			Hemoglobin Beta A Subunit	8.84	Metal, heme, iron, oxygen binding; Oxygen carrier
Transport Protein	√			Globin Domain Containing Protein	7.01	Metal, heme, iron, oxygen binding; Oxygen carrier
Structural Protein			√	Keratin Type II Cycloskeletal 6C	6.26	Involved in structural integrity of epithelial cells
Structural Protein			√	Keratin Type II Cycloskeletal 6A	6.93	Involved in wound healing and structural molecular activity
Structural Protein			√	Keratin 12	4.99	Involved in structural molecular activity
Structural Protein			√	Keratin 15	4.93	Involved in structural molecular activity
Others	√		√	Serpin Family F Member 1	6.58	Involved in negative regulation of angiogenesis and epithelial cell proliferation in prostate gland development; Positive regulation of neurogenesis
Others			√	IF Rod Domain Containing Protein	8.47	Involved in structural molecular activity
Others	√			C-Type Lectin Domain Family	6.81	Calcium, carbohydrate, heparin and kringle domain binding; Bone mineralization
Others	√			Src Homology 2 Domain Containing E	9.16	Recruited to tyrosine-phosphorylated sites, so functions as a regulatory module of intracellular signaling cascades

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
