# Peer review of "Characterization and Evaluation of Bone-Derived Nanoparticles as a Novel pH-Responsive Carrier for Delivery of Doxorubicin into Breast Cancer Cells"

_ijms, 2020, doi:10.3390/ijms21186721_

Round 1
Reviewer 1 Report
This is an interesting manuscript.
Please revise the text for errors.
GBA is reported as a doxorubicin delivery carrier for breast cancer cells. The system was well characterised regarding its physical and chemical characteristics. An enhancement of doxorubicin toxicity into cell lines was shown when linked to the GBA and compared with free doxorubicin.
A natural derivative of HA 19 or CA - the GBA might be an attractive material for DDS preparation and, thus,an improvement for scienctific community.
Author Response
Reviewer 1 Comments
- This is an interesting manuscript.
Please revise the text for errors.
GBA is reported as a doxorubicin delivery carrier for breast cancer cells. The system was well characterised regarding its physical and chemical characteristics. An enhancement of doxorubicin toxicity into cell lines was shown when linked to the GBA and compared with free doxorubicin.
A natural derivative of HA 19 or CA - the GBA might be an attractive material for DDS preparation and, thus,an improvement for scienctific community.
Ans: The manuscript has been revised by correcting the errors (highlighted).
Reviewer 2 Report
Summary: In this work, the use of goose bone ash is tested to deliver doxorubicin for cancer drug delivery. The drug delivery system is pH-dependent and has positive results for the intended application.
General comments: the manuscript is of relevance, the discussion of results is fair, but the extension of the text, the quality of the figures and the structure of the manuscript need improvement. The number of subsections and of figures is excessive and difficults the reading and the comprehension of the work. Major revisions are needes.
Specific points:
-P3L117, P4L148, P10L287: do not mention the name of the equipment used in the Results and Discussion section (place it in the Experimental section instead).
-Try to merge sections: e.g., sections 2.4 and 2.5, sections 2.11 and 2.12, and section 3.12 and 3.16.
-There are too many figures and some of them seems more relevant to be represented as tables (e.g., Fig. 1, 2, 6 and 7)
-Figure 3: reduce the size of the figures and remove the text from them since this information should be provided in the figure caption
-Figure 5: the values in the x- and y-axis are very small, please increase the font size.
-Figures 8 and 12: reorganize the figures to make them more compact and remove the text from them since this information should be provided in the figure caption
Author Response
Reviewer 2 Comments
- P3L117, P4L148, P10L287: do not mention the name of the equipment used in the Results and Discussion section (place it in the Experimental section instead).
Ans: The names of the equipment have been removed from the ‘Results and Discussion’ section (L117, L149, L285) and highlighted in green.
- Try to merge sections: e.g., sections 2.4 and 2.5, sections 2.11 and 2.12, and section 3.12 and 3.16.
Ans: The sections have been merged, e.g. section 2.4, 2.10 and 3.11) and highlighted.
- -There are too many figures and some of them seems more relevant to be represented as tables (e.g., Fig. 1, 2, 6 and 7)
Ans: Figures 1, 2, 6 and 7 have been presented as Tables 1, 2, 3 and 4 and highlighted.
- Figure 3: reduce the size of the figures and remove the text from them since this information should be provided in the figure caption
Ans: The size of the figure has been decreased (Fig. 1) and text has been removed from the Fig.
- Figure 5: the values in the x- and y-axis are very small, please increase the font size
Ans: The x and y axis have been manually formatted and the font size has been increased (Figure 3). The font size of the data obtained from the FTIR machine is fixed and cannot be increased. Hence, the font has been increased manually.
- Figures 8 and 12: reorganize the figures to make them more compact and remove the text from them since this information should be provided in the figure caption
Ans: Figures 8 and 12 have been reorganized (Figures 4 and 8) and texts have been removed from the figures. However, the names of the samples have not removed in order to avoid confusion.
Note: New data for cytotoxicity analysis of DOX-loaded GBA has been added for 25,000 cells/well where the toxic effect was more evident compared to 50,000 cells/well (Figure 11, Table 6), and highlighted.
Round 2
Reviewer 2 Report
In my opinion, the manuscript can be published in its current form with just a minor change:
- In Tables 3 and 4, please replace "Different pHs" by "pH"